# Macrophages in Oral Carcinomas: Relationship with Cancer Stem Cell Markers and PD-L1 Expression

**DOI:** 10.3390/cancers12071764

**Published:** 2020-07-02

**Authors:** Faustino J. Suárez-Sánchez, Paloma Lequerica-Fernández, Julián Suárez-Canto, Juan P. Rodrigo, Tania Rodriguez-Santamarta, Francisco Domínguez-Iglesias, Juana M. García-Pedrero, Juan C. de Vicente

**Affiliations:** 1Department of Pathology, Hospital Universitario de Cabueñes, 33394 Gijón, Asturias, Spain; faustinosuarezsanchez@gmail.com (F.J.S.-S.); juliansuarezcanto@gmail.com (J.S.-C.); fdoig59@yahoo.es (F.D.-I.); 2Department of Biochemistry, Hospital Universitario Central de Asturias (HUCA), 33011 Oviedo, Asturias, Spain; palomalequerica@gmail.com; 3Head and Neck Oncology UnitInstituto de Investigación Sanitaria del Principado de Asturias (ISPA), Instituto Universitario de Oncología del Principado de Asturias (IUOPA), Universidad de Oviedo, 33011 Oviedo, Asturias, Spain; jprodrigot@telefonica.net (J.P.R.); taniasantamarta@gmail.com (T.R.-S.); 4Department of Otolaryngology, Hospital Universitario Central de Asturias (HUCA), 33011 Oviedo, Asturias, Spain; 5Department of Surgery, University of Oviedo, 33006 Oviedo, Asturias, Spain; 6Ciber de Cáncer (CIBERONC), Instituto de Salud Carlos III, Av. Monforte de Lemos, 3-5. 28029 Madrid, Spain; 7Department of Oral and Maxillofacial Surgery, Hospital Universitario Central de Asturias (HUCA), 33011 Oviedo, Asturias, Spain

**Keywords:** oral squamous cell carcinoma, tumor-associated macrophages, tumor microenvironment, CD68, CD163, NANOG, PD-L1

## Abstract

Tumor-associated macrophages (TAMs) can be polarized into antitumoral M1 and protumoral and immunosuppressive M2 macrophages. This study investigated the clinical relevance of TAM infiltration in oral squamous cell carcinoma (OSCC), evaluating CD68 (M1 and M2 macrophage marker) and CD163 expression (M2 macrophage marker) in the tumor nests and surrounding stroma. Immunohistochemical analysis of both stromal/tumoral CD68^+^ and CD163^+^ TAMs was performed in paraffin-embedded tissue specimens from 125 OSCC patients, and correlated with clinical data. Potential relationships with the expression of cancer stem cell (CSC) markers and PD-L1 in the tumors were also assessed. Stromal CD163^+^ infiltration was significantly associated with the tumor location in the tongue, and stromal and tumoral CD68^+^ and CD163^+^-infiltrating TAMs were more abundant in nonsmokers and non-alcohol-drinkers. Strikingly, this study uncovers an inverse relationship between CD68^+^ and CD163^+^ TAMs and CSC marker expression (NANOG and SOX2) in OSCC. High infiltration of CD163^+^ TAMs in both tumor and stroma was strongly and significantly correlated with the absence of NANOG expression. Moreover, infiltration of both CD68^+^ and CD163^+^ TAMs was also significantly associated with high tumor expression of PD-L1. Our results suggest that there is a link between TAM infiltration and immune escape in OSCC.

## 1. Introduction

Oral squamous cell carcinoma (OSCC) is the most frequent cancer of the head and neck area [1]. In spite of recent advances in understanding the molecular biology, diagnosis, and treatment of the OSCC, including microsurgical reconstruction and advances in multimodal tumor therapy, the five-year survival rate has remained under 50% over the past 30 years [2], mainly due to local incontrollable recurrence or metastasis. In 1889, Paget described by first time the “seed and soil” hypothesis, wherein carcinomas induce changes in adjacent stromal cells, which contribute to neoplastic tissue invasion [3,4]. The tumor microenvironment (TME) is a complex system where tumor cells reprogram surrounding stromal cells in order to support tumorigenesis, cancer progression, and invasion of adjacent tissues [5]. TME is composed by a variety of stromal cells, such as fibroblasts, endothelial cells, pericytes, and immune cells. Unlike tumor cells, all these cells do not have mutations, although their behavior is modulated by several cytokines. Tumor-associated macrophages (TAMs) are the most abundant and important immune cells in the TME. In solid tumors, 5% to 40% of the tumor mass consists of macrophages [6]. Macrophages play a role as an interface between innate and acquired immunity, and can be polarized into M1 and M2 phenotypes, based on the expression of cytokines, receptors, and effector molecules [7]. Under physiological conditions, macrophages are polarized into proinflammatory and antitumor M1 phenotype; however, tumor cells can switch macrophages to alternatively activated M2 phenotype via several pathways (CCL-2, IL-1, IL-4, IL-6, IL-8, IL-10, IL-13, CSF-1, PD-1/PD-L1, CD47/SIRPα, TGFβ) [5,7,8]. M2 macrophages, in turn, secrete high levels of cytokines, chemokines, enzymes, and growth factors, such as VEGF, PDGF, TGF-β, FGF, uPA, and several matrix metalloproteinases, most of them encoded by genes that are transcriptional targets of NF-κB, NFAT, and STAT3 signaling pathways [9], thereby increasing inflammation as well as promoting tumor progression, immunosuppression, angiogenesis, migration, metastasis, and treatment resistance [5,7]. Furthermore, in vitro studies have shown that transformation from M1 to M2 can reverse their polarization depending on the chemokine stimuli [10]. TAMs are a mixed population of macrophages harboring both M1 and M2, but mainly composed by M2 macrophages, recruited and educated by cancer cells [6]. Immunohistochemistry has been widely employed to identify TAMs. Antibodies against CD68, a pan-macrophage marker, allows to identify all macrophages regardless of their phenotype, while the CD163 marker is a transmembrane scavenger receptor for haptoglobin–hemoglobin, highly expressed by M2 macrophages, and as such it has been widely recognized and used as marker for M2 macrophages [7]. CD163 has an essential role in eliminating hemoglobin–haptoglobin complexes and inducing the innate immune response [11].

Cancer stem cells (CSCs), or cancer-initiating cells, are a small population of cancer cells capable of self-renewal and multipotency. Macrophages are the most important ancillary cells regulating CSC activities [1]. However, the functional role of TAMs in OSCC is poorly understood, and the possible relationship between TAM infiltration and CSC phenotypes has not been investigated yet in oral carcinogenesis [11]. Among the numerous CSCs markers, NANOG and SOX2 have recently been identified as predictors of oral cancer risk in patients with oral precancerous lesions [12,13].

The host immune response is a key factor shaping the TME, and immune escape has been recognized as an important hallmark of cancer. Thus, compelling evidence indicates that immune tolerance in the TME is involved in tumor progression [8]. The axis of immune checkpoint inhibitors is represented by programmed-death ligand 1 (PD-L1) expressed in tumor cells, which binds to the programmed-death 1 receptor (PD-1) on activated T cells, delivering an inhibitory signal to those T cells that prevents tumor elimination from the immune system [14]. OSCC suppresses antitumor immunity by the induction of PD-L1 expression on M2 macrophages, causing T cell apoptosis [15].

The clinical relevance of macrophage subpopulations in cancer has not been clearly established. In several types of cancers, such as lymphoma, glioma, lung, gastric, thyroid, breast, and kidney cancers, higher CD163 expression on TAMs has been associated with a worse prognosis [8,16,17]. In head and neck squamous cell carcinoma, an association between CD163-positive macrophages and poor prognosis has been reported in univariate but not multivariate survival analysis [17]. Moreover, when the CD163 expression was analyzed by subgroups, it was revealed that the impact on patient survival was specifically and significantly associated with stromal, but not intratumoral CD163 expression [17].

In this study we aimed to investigate the clinical relevance and prognostic value of TAM infiltration in a homogeneous series of 125 OSCC patients, by means of immunohistochemical analysis of CD68 and CD163. Stromal/tumoral expression of CD68 and CD163 was evaluated, as well as possible relationships with the expression of CSCs markers and tumor PD-L1.

## 2. Materials and Methods

### 2.1. Ethical Approval

All procedures performed in studies involving human participants were in accordance with the ethical standards of the institutional and/or national research committee (Regional Ethical Committee from Principado de Asturias for the project PI19/01255 (136/19)) and with the 1964 Helsinki declaration and its later amendments or comparable ethical standards.

### 2.2. Patients and Tissue Specimens

This retrospective study was approved by the Institutional Ethics Committee of the Hospital Universitario Central de Asturias and by the Regional CEIC from Principado de Asturias. Written informed consent was obtained from the patients involved in the study. Tissue specimens from 125 patients with OSCC who underwent surgical treatment with curative purposes at the Hospital Universitario Central de Asturias between 1996 and 2007 were sourced from archival tissue blocks provided by the Principado de Asturias BioBank (PT17/0015/0023), integrated in the Spanish National Biobanks Network and they were processed following standard operating procedures. The histological diagnosis was confirmed by an experienced pathologist (FDI). Clinicopathologic data were collected from medical records. Patients with OSCC in this study (82 male and 43 female) ranged in age from 28 to 91 years (mean 58.69, standard deviation 14.34 years). Cancers were located in the tongue (*n* = 51, 41%), floor of the mouth (*n* = 37, 30%), gingiva (*n* = 22, 18%), buccal mucosa (*n* = 7, 6%), retromolar area (*n* = 6, 4%), and palate (*n* = 2, 1%). Smoking habit and alcohol consumption were seen in 84 (67%) and 69 (55%) patients, respectively. AJCC [18] stage T1 was detected in 27 (22%) patients, T2 in 54 (43%) patients, T3 in 16 (13%) patients, and T4 in 28 (22%) patients. Stage pN0 was observed in 76 (61%), pN1 in 25 (20%), and pN2 in 24 (19%) patients. According to overall AJCC stages, 20 (16%) had stage I, 32 (26%) had stage II, 26 (20%) had stage III, and finally, 47 (38%) had stage IV. Regarding histopathologic degree of differentiation, 80 (64%) OSCCs were well-differentiated, 41 (33%) moderately, and 4 (3%) poorly-differentiated. During the follow-up period (6 to 230 months) 54 (43%) patients had local or regional recurrence, 19 (15%) suffer from a second primary cancer, and 53 (42%) died of OSCC.

### 2.3. Tissue Microarray (TMA) Construction

Three morphological representative areas were selected from each individual paraffin tumor block, including both the invasive border as well as the center of tumor sheets or islands without necrotic areas. In addition, each TMA contained morphologically normal oral mucosa samples from nononcological patients undergoing oral surgery as internal negative controls. In order to check the histopathological diagnosis and the adequacy of tissue sampling, a section from each microarray was stained with hematoxylin and eosin and examined by light microscopy.

### 2.4. Immunohistochemistry (IHC)

The TMAs were cut into 3 μm sections and dried on Flex IHC microscope slides (DakoCytomation, Glostrup, Denmark). The sections were deparaffinized in xylene and rehydrated through a graded alcohol series. Antigen retrieval was performed by heating the sections with Envision Flex Target Retrieval solution, high pH (Dako, Glostrup, Denmark). Staining was done at room temperature on an automatic staining workstation (Dako Autostainer Plus, Dako). The following primary antibodies were used: anti-CD68 (Agilent-Dako, clone KP1, prediluted), anti-CD163 (Biocare Medical, Pacheco, CA, USA; clone 10D6, 1:100 dilution), mouse monoclonal PD-L1 antibody (clone 22C3, PD-L1 IHC 22C3 pharmDx, Dako SK006; 1:200 dilution), NANOG (D73G4) XP^®^ rabbit monoclonal antibody (Cell Signaling technology, Inc., Leiden, The Netherlands; 1:200 dilution), and anti-SOX2 rabbit polyclonal antibody (AB5603, Merck Millipore, Darmstadt, Germany; 1:1000 dilution) by using the Dako EnVision Flex + Visualization System (Dako Autostainer) and diaminobenzidine chromogen as substrate. Negative controls were prepared by omitting the primary antibody. Positive controls were prepared using appropriate positive control slides. Counterstaining with hematoxylin was the final step. The IHC results were independently evaluated by four observers (FDI, FJSS, JPR, and JMG-P) blinded to clinical data. The number of CD163-positive cells was counted in each 1 mm^2^ area from three independent high-power representative microscopic fields (HPFs, 400×; 0.0625 μm^2^), in both the tumor nests and the surrounding stroma. In the survival analysis, all the specimens were divided into low and high groups on the basis of the number of positive cells/mm^2^, using cut-off values of 25, 50 (median), and 75 percentiles, both for CD68- and CD163-positive cells.

PD-L1 immunostainings was semiquantitatively scored into five different categories according to the percentage of stained tumor cells (0, negative; 1, 1–10%; 2, 10–25%; 3, 26–50%; and 4, more than 50% of stained cells). Since PD-L1 expression in more than 10% of tumor cells was associated with a poorer survival [19], this was established as a cut-off point for subsequent analyses. Accordingly, for analytical purposes, OSCC patients were dichotomized into two groups: relevant versus nonrelevant PD-L1 expression, based on the established cut-off point of 10% of stained tumor cells. NANOG expression was scored as negative (absence of staining, score 0), weak to moderate (some cytoplasmic staining in dysplastic areas, score 1), and strong protein expression (intense and homogeneous cytoplasmic staining in dysplastic areas, score 2), and SOX2 expression was evaluated as the percentage of tumor cells with positive nuclear staining, following previously described methods [12,13]. For statistical purposes, NANOG staining was dichotomized as negative (score 0) versus positive expression (scores 1–2). SOX2 staining scores were classified as negative or positive expression on the basis of values below or above the median value of 10%, respectively. A sample of testicular seminoma (a tumor known to express NANOG and SOX2) was used as a positive control.

### 2.5. Statistical Analysis

χ^2^ and Fisher’s exact tests were used for comparison between categorical variables. Comparison of mean values was performed with a *t* test. Disease-specific survival (DSS) was determined as the time from the initial treatment completion to the date of death due to the tumor or the presence of a nontreatable recurrence. For time-to-event analysis, survival curves were plotted using the Kaplan–Meier method. Differences between survival times of subgroups of patients were compared by using Mantel’s log-rank test. All *p* values were based on the two-sided statistical analysis, and a *p* value less than 0.05 was considered as statistically significant. All statistical analyses were performed using SPSS version 18 (IBM Co., Armonk, NY, USA).

## 3. Results

### 3.1. Expression of CD68 and CD163 in OSCC Tissue Specimens

To investigate the density and distribution of TAMs in OSCC, CD68 and CD163 expression was analyzed by immunohistochemistry in 125 OSCC patients, using tissue microarrays. The mean number of CD68^+^ macrophages in the tumor nests and the surrounding stroma was 51.11 ± 45.08 per mm^2^ (range: 0.00 to 194.00) and 122.72 ± 82.49 per mm^2^ (range: 9.67 to 488.67), respectively (Figure 1). CD163 staining was primarily detected in the cytoplasm of macrophages (Figure 2). The mean CD163 expression scores in the tumor nests and the tumor stroma were 31.116 ± 28.91 (range: 0.00 to 187.33) and 168.146 ± 97.591 (range: 14.33 to 476.67) cells per mm^2^, respectively.

### 3.2. Associations between CD68 and CD163 Expression and Clinicopathological Variables

CD68^+^ TAM infiltration (either categorized as mean number of positive cells or staining intensity) showed no significant associations with any of the clinicopathological parameters studied (age, gender, tobacco or alcohol consumption, TNM classification, differentiation grade, tumor location, recurrence or second primary tumors) (Table 1).

Stromal CD163^+^ infiltration was significantly associated with the tumor location, and higher in the tongue (*p* = 0.02) (Table 2). Even though no other significant associations were found with all the remaining clinicopathological variables, it is worth noting that higher stromal and tumoral CD163^+^-infiltrating cells were observed in nonsmokers and non-alcohol-drinkers. Similarly, stromal and tumoral CD68^+^ infiltration was higher in those patients without tobacco/alcohol habits (Table 1). Moreover, higher stromal CD68^+^ and stromal CD163^+^-infiltrating cells were observed in bigger tumors (T3–T4), advanced III–IV stages and moderate/poorly differentiated tumors, although differences did not attain statistical significance (Table 1).

### 3.3. Correlations between the Expression of CD68, CD163, and CSC Markers

The mean number of CD68^+^-TAM infiltration in both the tumor and the surrounding stroma was higher in tumors harboring negative expression of NANOG and SOX2, although these differences did not reach statistical significance (Table 2).

Concordantly, the mean number of stromal/tumoral CD163^+^ cells was also higher in tumors with negative expression of NANOG and SOX2. Noteworthy, CD163^+^ TAM infiltration was inversely and significantly correlated with NANOG expression, in both the stroma (*p* = 0.001) and tumor nests (*p* = 0.03) (Table 2).

### 3.4. Correlation between TAM Infiltration and PD-L1 Expression

Positive PD-L1 expression in more than 10% of tumor cells was found in 18 (14.4%) in our series of OSCC samples, as previously reported [19]. CD68^+^-infiltrating cells in both the stroma and the tumor nests was significantly associated with PD-L1-positive tumors (*p* = 0.04 and *p* < 0.0001, respectively; Table 3). Similarly, stromal and tumoral CD163^+^ TAM infiltration was consistently and significantly correlated with positive PD-L1 expression (above 10% of tumor cells) (*p* = 0.01 and *p* < 0.0001, respectively; Table 3).

### 3.5. Impact of TAM Infiltration on the Survival of OSCC Patients

Follow-up information was available for 121 patients with OSCC, ranging from 6 to 230 months with a mean of 74.10 (SD: 57.08) and a median of 61 months. The median follow-up for all patients was 61 months, and for survivors was 113 months. At the end of this study, 17 (14%) patients were lost during the follow-up period, 53 (44%) patients were alive and free of recurrence, and finally, 51 (42%) of them died of cancer or showed a nontreatable recurrence.

Correlations between TAM infiltration (as mean numbers of CD68^+^ and CD163^+^ cells) and the disease-specific survival of OSCC patients were assessed by using different cut-off values (based on 25, 50 (median), and 75 percentiles), and survival rates calculated using the Kaplan–Meier method. No significant associations were observed between the number of CD68^+^ and CD163^+^-infiltrating cells and patient survival regardless the different cut-off values used (Appendix A).

## 4. Discussion

We herein investigated the density and tissue distribution of TAMs infiltration by immunohistochemically evaluating CD68 and CD163 staining in 125 OSCC samples, and potential relationships with the expression of NANOG and SOX2 as major CSC markers, and PD-L1 expression. CD68 is a marker of M1 and M2 activated TAMs [17]. The analysis of CD68^+^ TAM infiltration both in tumor nests and the surrounding stroma showed no associations with clinicopathological variables and survival in our cohort of 125 OSCC patients. Lin et al. [20] studied TAMs infiltration by assessing CD68 expression using immunohistochemistry in 84 laryngeal carcinomas found to significantly correlate with tumor recurrence and poor survival. Contrasting this, Troiano et al. [17] recently conducted a meta-analysis in HNSCC, and reported no association between tumor or stromal expression of CD68^+^ macrophages and survival. In the present study, further analysis of CD163 expression, a highly specific marker for M2 macrophages, was also performed. M2 macrophages infiltrating the tumor nests and the tumor stroma were separately scored. The number of CD163^+^-infiltrating TAMs was not significantly associated with any clinicopathological parameters or prognosis in none of these two compartments. These results were consistent with those reported in other OSCC studies [2,16,21]. However, it has also been reported that a high number of CD163^+^ macrophages was significantly correlated with a worse survival [5,17,22,23]. Interestingly, in some studies, only stromal expression of CD163 resulted to be significantly associated with survival, whereas no significance was detected for intratumoral expression [17]. Hence, the association between the expression of CD163 and clinical prognosis in OSCC patients remains controversial. Surprisingly, high density of TAMs in colorectal cancer seems to be associated with a better survival, while no significant correlation was observed with survival in some esophageal cancer patients [24]. The explanation to these inconsistencies may be related to differences in tumor biology, the methodologies used to assess macrophage infiltration (e.g., systemic counting or hot spot counting, counting tumoral or stromal macrophages, or both), immunohistochemical evaluation, as well as differences in the number of clinical cases among studies, tumor extension, tumor stage, and the as of yet not well understood role of M2-macrophages in tumor biology.

Cancer stem cells (CSCs) are defined as a small subpopulation of cells in the tumors that possess the ability to initiate neoplasms and sustain tumor self-renewal. It has been suggested that CSCs rely on a TME niche, demonstrated using knockdown experiments that CD163 promoted cell proliferation and stemness [11], thereby contributing to tumorigenesis. This could be, at least partially explained, because CD163 contributes to regulating the transcription of cyclin D1, CD133, ALDH1A1, NANOG, and OCT4 [11]. However, here, we consistently found higher infiltration of CD163^+^ TAM in tumors with negative expression of the CSC markers NANOG and SOX2. In particular, stromal and tumoral CD163^+^ TAMs were inversely and significantly associated with NANOG expression, thereby suggesting an inverse correlation between TAM infiltration and stemness in OSCC. This relationship could suggest a potential role for the CSC niche in immune evasion and OSCC progression, plausibly through paracrine signals to control TAMs infiltration in the tumor microenvironment. NANOG is a transcription factor that is a key regulator of pluripotency in stratified epithelia [25], and mediates tumor cell proliferation, epithelial–mesenchymal transition, and escape from immune system [26]. A feedback system between CSCs and TAMs has been described, based on the recruitment of TAMs through the blood vasculature, and chemokine release by the TAMs to maintain CSCs quiescence [1]. TAMs have been traditionally considered as blood monocytes recruited from the tumor vasculature by tumor-derived signals [27], but more recently it has been shown that most tissue macrophages arise from yolk sac progenitors, with some exceptions such as those from intestines [28]. This fact raises the possibility that TAMs could have a CSCs origin in the tumor microenvironment [29]. NANOG and SOX2 maintain embryonic stem cells (ESCs) properties [30,31], as well as CSC. We herein unveil an unprecedented relationship between TAM infiltration and the oral CSC niche, perhaps reciprocally regulated by paracrine signals. Nevertheless, the precise role of TAMs in regulating CSCs and the molecular mechanisms underlying these interactions in the context of OSCC are not known. In the interplay between CSCs and TAMs, a putative role for the signal transducer and activator of transcription 3 (STAT3) pathway has been demonstrated in breast cancer models via a novel paracrine EGFR/Stat3/Sox-2 axis [32]. In the TME, hypoxia stimulates the expression of interleukin-6, which in turn induces STAT3 in CSCs, and M2 macrophage polarization [33,34]. While the ability of CSCs to induce macrophage polarization is well-established, the reciprocal effect of macrophages on the CSC phenotype remains unclear. Nusblat et al. [34] showed that in the presence of conditioned media from M2 macrophages, CSCs increased migration and collagen degradation capacities. In addition, CSCs may exert a strong immune suppressive effect by suppressing the Major Histocompatibility Complex (MHC) by macrophages in the TME, releasing the exosomal miRNAs miR-9 and miR-21, or by secreting VEGF that plays a pivotal role in tumor angiogenesis and suppression of T-cell maturation [35]. Mazzoldi et al. [36] found that, the c-Kit ligand Stem Cell Factor (SCF), produced by M2 macrophages, may favor the immune escape of tumor cells in ovarian cancer. Similarly, ovarian CSCs and macrophages reciprocally interact through the WNT/β-catenin signaling, thereby contributing to tumorigenesis, invasiveness, and immune suppression [37]. Our study demonstrated for the first time a strong inverse correlation between both stromal and tumoral CD163^+^ TAMs and NANOG expression. According to these findings, we hypothesize that at least M2 TAM infiltration does not seem to be related to CSC niche maintenance, and perhaps could play a fundamental role in the tumor immune evasion. Nevertheless, further studies are needed to fully depict the functions, underlying mechanisms, and specific contexts linking TAMs and stemness.

Recent studies reported that CD163^+^ macrophages were associated with PD-L1 expression on tumor cells in several human cancers [38,39,40]. In good agreement, we found that both stromal and tumor-infiltrating M2 macrophages were associated with PD-L1 expression, which was previously associated with poor prognosis in OSCC [19]. Experimental studies have reported that tumor cells can induce M2 macrophage phenotype increasing the expression of PD-L1 [41]. In fact, PD-L1 expression is induced after exposure to interferon-γ (IFN-γ) released by T effector cells, as well as after other signals such as TNF-α, VEGF, and CXCL8, ultimately promoting lung cancer progression [42,43,44,45]. Yagyuu et al. [46] revealed that the increase in the number of subepithelial CD163^+^ cells was significantly correlated with the presence of high-grade oral epithelial dysplasia, and the coexpression of PD-L1 and CD163 was found in 16.6% of subepithelial cells. In hepatocellular carcinomas [47], in HPV-associated tonsil squamous cell carcinomas [48], and in gastric adenocarcinoma [38], the infiltration rates of CD163-positive cells were significantly higher in tumors that expressed PD-L1, suggesting that M2 macrophage infiltration could be used as a predictive marker for PD-L1 expression. Nevertheless, the relationship between M2 TAM and PD-L1 has not been completely clarified. Tyro3, Axl, and Mertk, collectively called TAM receptors, can activate the expression of PD-L1 in tumor cells [49], and additionally, IFN-γ secreted by inflammatory cells in the TME is associated with macrophage differentiation. Interestingly, IFN-γ induces the expression of PD-L1 in tumor cells and also the protein kinase D isoform 2 (PKD2), an important regulator of PD-L1 in OSCC [50]. Furthermore, Fujita et al. [2] reported the possibility that IL-8 produced by cancer cells stimulates CD163^+^ M2 macrophages to produce IL-10, which, in turn, leads to the phosphorylation of STAT3, and then IL-10/STAT3 signaling induces PD-L1 overexpression [51]. STAT3 signaling is constitutively activated in various tumors and is also involved in the regulation of monocyte-chemotactic cytokines [9]. We speculate that this relationship between CD163 and PD-L1 expression could be the link between inflammation and immune escape in oral carcinogenesis. OSCC can be promoted by chronic inflammation, such as that which occurs in periodontitis [52]. Furthermore, it has been shown that a trauma, including an incisional biopsy, can provide a microenvironmental stimulus that affects macrophage polarization influencing tumor biology, leading to a worse prognosis and increasing the risk of developing lymph node metastasis in OSCC [53]. Weber et al. [53] demonstrated a shift in macrophage polarization towards the M2 phenotype in the time interval between diagnostic incisional biopsy and definitive tumor resection in OSCC.

## 5. Limitations

A possible limitation of this study is that we used CD163 as an M2 macrophage marker, which is the most currently used in the literature [5]; however, it is worth noting that M1 and M2 phenotypes are the extremes of a continuous spectrum of macrophage polarization [54], and macrophages display high plasticity in response to different stimuli. Consequently, some authors [55] consider that the single staining with CD163 is not sufficient for allocating macrophages towards M2 polarization. Another potential limitation is that this is a retrospective study that cannot exclude potential selection bias.

## 6. Conclusions

This study thoroughly investigated the clinical relevance of TAM infiltration in OSCC, jointly evaluating CD68 and CD163 expression in both the tumor nests and surrounding stroma. Results consistently showed that stromal/tumoral TAM infiltration had no impact on patient prognosis and disease progression/outcome. Interestingly, this study uncovers an inverse relationship between CD68^+^ and CD163^+^ TAMs and CSC marker expression in OSCC. In particular, high infiltration of CD163^+^ TAMs was strongly and significantly correlated with the absence of NANOG expression. Moreover, infiltration of both CD68^+^ and CD163^+^ TAMs was also significantly associated with a high expression of PD-L1 in the tumors, a PD-1 ligand that negatively regulates local immunity, suggesting a link between TAM infiltration and immune escape in OSCC. A more profound elucidation of the role of TAMs could lead to the identification of novel macrophage-therapeutic targets in OSCC.

## Figures and Tables

**Figure 1 cancers-12-01764-f001:**
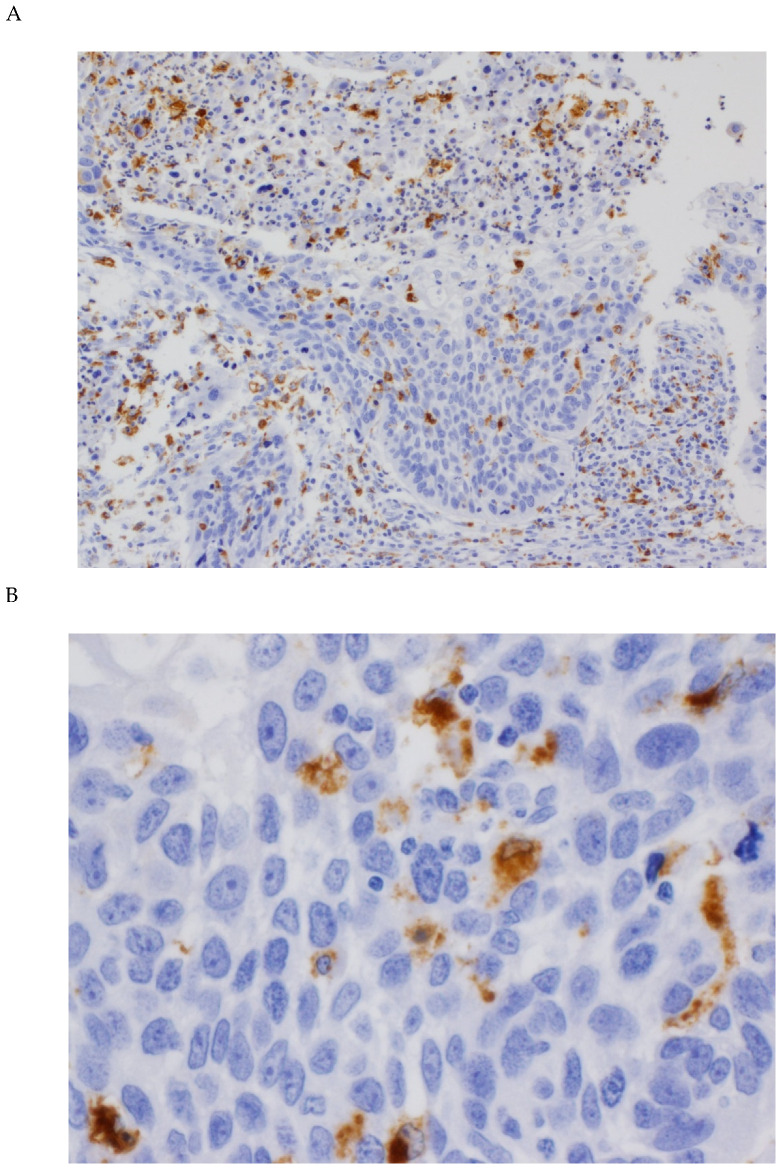
Immunohistochemical analysis of CD68 expression in OSCC tissue specimens. (**A**) Representative example of positive CD68 expression in tumor-associated macrophages (TAMs) both in tumor nests and in the stroma (original magnification ×100). (**B**) CD68 cytoplasmic staining in cells identified as TAMs (original magnification ×400).

**Figure 2 cancers-12-01764-f002:**
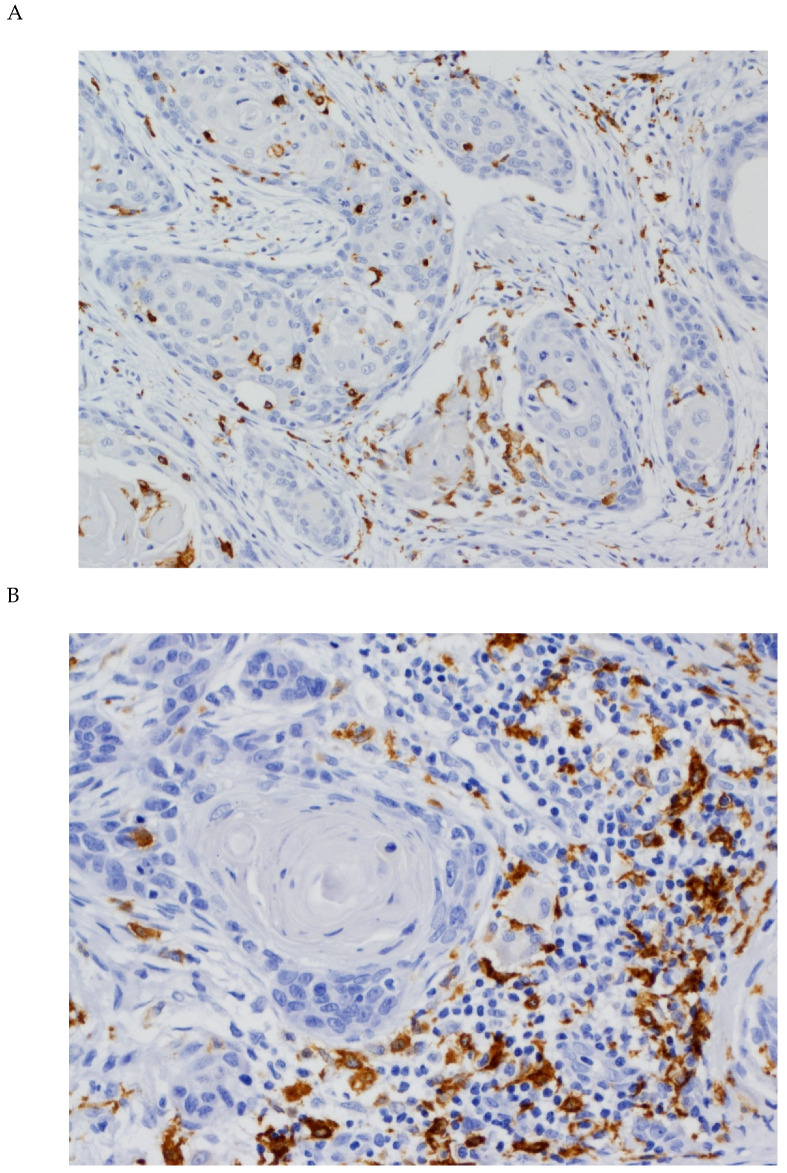
Immunohistochemical analysis of CD163 expression in OSCC tissue specimens (**A**) CD163 expression both in stromal and tumoral TAMs (original magnification ×100). (**B**) CD163 expression in stromal TAMs (original magnification ×400).

**Table 1 cancers-12-01764-t001:** Correlations of CD68^+^ and CD163^+^ macrophages with clinicopathological parameters in oral squamous cell carcinoma (OSCC) patients.

Variable	Number (%)	Stromal CD68 Mean (SD)	*p*	Tumoral CD68 Mean (SD)	*p*	Stromal CD163 Mean (SD)	*p*	Tumoral CD163 Mean (SD)	*p*
**Age (Years)**			0.78		0.56		0.89		0.28
**<65**	77 (62)	124.34 (88.26)		49.08 (46.59)		169.10 (105.50)		28.91 (30.74)	
**≥65**	48 (38)	120.13 (73.13)		52.80 (46.75)		166.60 (84.40)		34.64 (25.62)	
**Gender**			0.76		0.82		0.45		0.82
**Female**	43 (34)	125.78 (96.54)		47.93 (29.74)		159.06 (90.40)		30.31 (29.74)	
**Male**	82 (66)	121.12 (74.69)		31.53 (28.64)		172.90 (101.36)		31.53 (28.64)	
**Tobacco**			0.28		0.05		0.25		0.17
**No**	41 (33)	134.04 (80.06)		62.22 (50.03)		182.44 (96.12)		36.12 (28.52)	
**Yes**	84 (67)	117.20 (83.57)		45.62 (41.67)		161.16 (98.10)		28.67 (28.96)	
**Alcohol Consumption**			0.10		0.13		0.32		0.10
**No**	56 (45)	136.11 (85.67)		57.72 (46.32)		177.82 (91.55)		35.77 (25.93)	
**Yes**	69 (55)	111.86 (78.77)		45.67 (43.63)		160.28 (102.21)		27.33 (30.79)	
**T**			0.74		0.25		0.40		0.44
**T1 + 2**	81 (65)	124.53 (80.99)		47.44 (40.39)		173.53 (100.19)		29.50 (25.21)	
**T3 + 4**	44 (35)	119.40 (86.04)		57.79 (52.41)		158.22 (92.91)		34.08 (34.84)	
**N**			0.92		0.64		0.32		0.72
**N0**	76 (61)	123.26 (80.08)		52.62 (43.72)		160.67 (84.64)		31.83 (32.04)	
**N+**	49 (39)	121.89 (86.94)		48.80 (47.47)		179.72 (114.82)		29.99 (23.53)	
**Stage**			0.75		0.25		0.83		0.19
**I + II**	52 (42)	125.50 (87.57)		45.62 (40.87)		166.00 (93.61)		27.16 (25.73)	
**III + IV**	73 (58)	120.74 (79.24)		54.95 (47.71)		169.67 (100.94)		33.92 (30.84)	
**Grade**			0.81		0.13		0.62		0.34
**Well**	80 (64)	124.05 (86.15)		46.05 (38.52)		164.89 (97.59)		29.28 (28.68)	
**Moderate + Poor**	45 (36)	120.37 (76.44)		60.00 (54.08)		173.92 (98.41)		34.37 (29.35)	
**Site**			0.21		0.94		0.02		0.24
**Tongue**	51 (41)	133.69 (86.47)		51.46 (46.43)		193.00 (117.24)		27.50 (22.38)	
**Rest**	74 (59)	115.16 (79.35)		50.88 (44.47)		151.01 (77.67)		33.60 (32.58)	
**Second Primary Tumor**			0.43		0.98		0.90		0.22
**No**	106 (85)	125.19 (87.02)		51.14 (44.08)		167.69 (98.51)		32.43 (29.68)	
**Yes**	19 (15)	108.98 (49.70)		50.93 (52.02)		170.68 (94.83)		23.73 (23.52)	

**Table 2 cancers-12-01764-t002:** Correlation between CD68^+^ and CD163^+^ macrophages and the expression of cancer stem cell (CSC) markers in OSCC.

Factor	SOX2 Expression		*p*	NANOG Expression		*p*
	Negative	Positive		Negative	Positive	
	(*n* = 72, 60%)	(*n* = 49, 40%)		(*n* = 83, 68%)	(*n* = 39, 32%)	
**Stromal CD68 (Mean, SD)**	129.55 (85.06)	113.71 (75.68)	0.29	133.97 (89.65)	103.43 (61.67)	0.05
**Tumoral CD68 (Mean, SD)**	54.00 (44.79)	50.87 (45.56)	0.70	55.85 (47.50)	42.05 (39.40)	0.11
**Stromal CD163 (Mean, SD)**	179.84 (98.14)	155.55 (95.18)	0.17	183.79 (106.56)	133.53 (55.78)	0.001
**Tumoral CD163 (Mean, SD)**	33.65 (31.25)	29.82 (25.07)	0.47	34.42 (31.64)	24.16 (20.98)	0.03

**Table 3 cancers-12-01764-t003:** Correlation between CD68^+^ and CD163^+^ macrophages and PD-L1 expression (clone 22C3) in OSCC.

Factor	Tumor PD-L1		*p*
	≤10%	>10%	
	(*n* = 104, 85%)	(*n* = 18, 15%)	
**Stromal CD68 (Mean, SD)**	114.31 (66.89)	183.59 (130.68)	0.04
**Tumoral CD68 (Mean, SD)**	46.39 (40.98)	86.03 (52.99)	<0.0001
**Stromal CD163 (Mean, SD)**	161.28 (91.29)	224.83 (114.23)	0.01
**Tumoral CD163 (Mean, SD)**	27.28 (23.66)	58.20 (41.02)	<0.0001

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
