# Peer review of "Macrophages in Oral Carcinomas: Relationship with Cancer Stem Cell Markers and PD-L1 Expression"

_cancers, 2020, doi:10.3390/cancers12071764_

Round 1

Reviewer 1 Report

  1. In the article “Oral Oncol. 2019 Jun;93:66-75. Prognostic significance of CD68+and CD163+ tumor associated macrophages in head and neck squamous cell carcinoma: A systematic review and meta-analysis.”, Troiano et al has reported that High stromal expression of CD163+ TAMs correlated with both poor overall survival (HR, 2.26; 95% CI: [1.47, 3.47]; P < 0.001) and progression-free survival (HR, 2.29; 95% CI: [1.11, 4.71]. Therefore, the novelty of this manuscript seems not enough.

  1. In this manuscript, authors only showed the clinical correlation results. Authors did not present any experiments to explain the relationship between TAM and OSCC. How do TAM promote OSCC progression? How do OSCC promote recruitment of TAM? Authors should provide experiments to further validate the clinical findings.

  1. Author described the follow-up period was 6 to 230 months. Please describe the median follow-up for all patients and survivors in the manuscript.

  1. In the method, authors described that Disease-specific survival (DSS) was determined as the time from the initial treatment completion to the date of death due to the tumor or the presence of a non-treatable recurrence. The definition seems incorrect. The presence of a non-treatable recurrence is one of the criteria for disease-free survival, not disease-specific survival. The definition and calculation of Disease-specific survival and disease-free survival are different. Please clarify.

  1. In the method, 19 (15%) suffer from a second primary cancer. What kind of cancers was noted in these 19 patients? If the patient had tongue cancer initially, he developed the har palate cancer later. Both tongue cancer and hear palate cancer are OSCC. Does this patient belong to recurrence or secondary primary cancer ?

  1. In the introduction, authors described that OSCC suppresses antitumor immunity by the induction of PD-L1 expression on M2 macrophages, causing T cell apoptosis. In this manuscript, authors have performed PD-L1 staining. But, authors did not analyze PD-L1 expression on macrophages in these 125 OSCC. Could authors analyze PD-L1 expression on macrophages and correlate them with other IHC markers and clinicopathological parameters?

  1. In the method, authors described Disease-specific survival (DSS). However, in table 4 and 5, disease-free survival was showed. They are different. Please clarify

  1. Except Disease-specific survival and disease-free survival, it is better to show the overall survival

  1. In this manuscript, author showed the mean number of stromal/tumoral CD163+ cells were also higher in tumors with negative expression of NANOG and SOX2. Noteworthy, CD163 + TAM infiltration was inversely and significantly correlated with NANOG expression, in both the stroma (p = 0.001) and tumor nests (p = 0.03). Please described the more detailed rationale to correlated TAM with CSC markers. What is the significance? I think it is more significant if authors can use some in vitro experiments to validate this clinical finding.

  1. In table 1, “recurrence” should not be put in table 1. “Recurrence” is time-dependent. It should be presented in disease-free survival using Kaplan-Meier survival curve.

Author Response

  1. In the article “Oral Oncol. 2019 Jun;93:66-75. Prognostic significance of CD68+and CD163+tumor associated macrophages in head and neck squamous cell carcinoma: A systematic review and meta-analysis.”, Troiano et al has reported that High stromal expression of CD163+ TAMs correlated with both poor overall survival (HR, 2.26; 95% CI: [1.47, 3.47]; P < 0.001) and progression-free survival (HR, 2.29; 95% CI: [1.11, 4.71]. Therefore, the novelty of this manuscript seems not enough.

The study by Troiano et al. has been included in the Discussion of our manuscript. These authors actually found that high expression of CD163+ TAM was correlated with worse survival in head and neck squamous cell carcinoma (SCCHN) patients (HR,2.26; 95% CI: [1.47, 3.47]; P < 0.001); however, subgroup analysis revealed that these results were dependent upon the type of survival analysis performed. Specifically, the association between CD163+ TAM and OS showed statistical significance only in univariate (HR, 2.52; 95% CI: [1.60, 3.97]; P < 0.001) but not multivariate (HR, 0.99; 95% CI: [0.28, 3.50]; P = 0.99) survival analysis. In addition, when localization of TAM infiltration was considered, a differential impact on prognosis was also observed. Thus, the association was significant for stromal (HR, 2.46; 95% CI: [1.43, 4.24]; P < 0.001), but not for intratumoral TAM (HR, 1.93; 95% CI: [0.81, 4.62]; P = 0.14) (Fig. 5). In the study by Troiano et al., a meta-analysis for DFS was not performed because of the absence of available data. Here we thoroughly assessed the prognostic significance of TAM infiltration using a large homogeneous cohort of surgically treated OSCC patients. To this purpose, two different markers CD68 and CD163 were meticulously evaluated, thereby separately considering and scoring both the stromal as well as the intratumoral TAM infiltration and on each case evaluating three different cutoff points depending of the corresponding 25, 50 and 75 percentiles. Even though the prognostic relevance of TAM infiltration has been previously investigated, we truly think that the unique characteristics in terms of the study design and our findings bring together important novelty and added value to the field. Moreover, this study also investigates the relationships of TAM infiltration with the expression of PD-L1 and the cancer stem cell (CSC) markers NANOG and SOX2 (two well-known regulators of stemness and CSC properties) Regarding the novelty of this manuscript, we did not only study the prognostic relevance of CD68 and CD163 markers, but also their relationship with. As far as we know, our study is the first to demonstrate an unprecedented strong inverse correlation between CD68+ and CD163+TAMs and CSC marker expression in OSCC. In particular, quite consistently both stromal and tumoral CD163+ TAMs was inversely and significantly correlated with NANOG expression. We believe that this is a novel and potentially relevant finding.

  1. In this manuscript, authors only showed the clinical correlation results. Authors did not present any experiments to explain the relationship between TAM and OSCC. How do TAM promote OSCC progression? How do OSCC promote recruitment of TAM? Authors should provide experiments to further validate the clinical findings.

As stated, the aim of this study was indeed to investigate the clinical relevance and prognostic value of TAM infiltration in a homogeneous series of 125 OSCC patients, by means of immunohistochemical analysis of CD68 and CD163. Stromal/tumoral expression of CD68 and CD163 was evaluated, as well as possible relationships with the expression of CSCs markers and tumor PD-L1. Please take into the consideration that the molecular basis of these clinical findings as well as the nature of the relationships between TAM infiltration and the CSC niche are beyond the scope of our study and also outside the time frame of this revision.

  1. Author described the follow-up period was 6 to 230 months. Please describe the median follow-up for all patients and survivors in the manuscript.

The median follow-up for all patients was 61 months, and for survivors was 113 months. This information has now been included in section 3.5, according to the reviewer’s recommendation.

  1. In the method, authors described that Disease-specific survival (DSS) was determined as the time from the initial treatment completion to the date of death due to the tumor or the presence of a non-treatable recurrence. The definition seems incorrect. The presence of a non-treatable recurrence is one of the criteria for disease-free survival, not disease-specific survival. The definition and calculation of Disease-specific survival and disease-free survival are different. Please clarify.

Disease-free survival (DFS) can be calculated as the time from the initial treatment of a cancer to the date of first cancer recurrence or death.

Disease-specific survival (DSS) can be defined as the time period from the primary treatment of a cancer to the time of death. Patients who died from causes other than the cancer being studied are not counted in this measurement.

The overall survival (OS) (or total mortality) is defined as the time from primary treatment (surgery, chemotherapy, radiotherapy) to the date of death of any cause or last follow-up.

Here we assessed the time from the initial treatment (surgery in all cases) of OSCC patients to the time of death or the presence of a non-treatable recurrence (not the presence of a first or second recurrence if this was treatable) that produced the death of the patient in two or three weeks. Consequently, we think that this definition matches better with DSS than with DFS. This is the definition of DSS that we extensively used in our previous papers.

  1. In the method, 19 (15%) suffer from a second primary cancer. What kind of cancers was noted in these 19 patients? If the patient had tongue cancer initially, he developed the har palate cancer later. Both tongue cancer and hear palate cancer are OSCC. Does this patient belong to recurrence or secondary primary cancer?

The criteria used to diagnose multiple primary carcinomas was originally described by Warren and Gates (Warren S, Gates DC. Multiple primary malignant tumors: a survey of the literature. Am J Cancer 1932; 16:1358-414), and later modified by Hong et al. (Hong WK, Lippman SC, Itri L, Karp D, Lee JS, Byers RM, et al. Prevention of second primary tumors with isotretinoin in squamous cell carcinoma of the head and neck. N Engl J Med 1990;323(12):795-801) as follows:

(1) each neoplasm must be geographically separate and distinct (if the intervening mucosa demonstrates dysplasia, it is considered a multicentric primary), and (2) the possibility that the second primary represents a metastasis or a local relapse must be excluded.

A second primary has to be separated from the first by at least 2 cm of normal epithelium or if this condition is not met, had to occur at least 3 years after the first diagnosis. Synchronous carcinomas has been defined as second neoplasms at the same time or within a 6-month period of the primary lesion. After this period they were considered metachronous neoplasms.

In 19 patients included in this study a second primary cancer developed in the oral cavity in different sites than the initial or index tumors.

If the patient had tongue cancer initially, he developed the har palate cancer later. Both tongue cancer and hear palate cancer are OSCC. Does this patient belong to recurrence or secondary primary cancer?

If a patient has a squamous cell carcinoma located in the tongue, and after treatment, he/she developed a new squamous cell carcinoma in the same location of the primary tumor, and other squamous cell carcinoma in the hard palate, the tongue cancer is considered a recurrence and the other one a second primary neoplasm (multiple primary carcinomas).

  1. In the introduction, authors described that OSCC suppresses antitumor immunity by the induction of PD-L1 expression on M2 macrophages, causing T cell apoptosis. In this manuscript, authors have performed PD-L1 staining. But, authors did not analyze PD-L1 expression on macrophages in these 125 OSCC. Could authors analyze PD-L1 expression on macrophages and correlate them with other IHC markers and clinicopathological parameters?

Programmed cell death protein 1 (PD-1) is an immune checkpoint receptor that is upregulated on activated T cells for the induction of immune tolerance. Tumor cells frequently overexpress the ligand for PD-1, programmed cell death ligand 1 (PD-L1), facilitating their escape from the immune system. Gordon et al. (Nature 2017;545(7655): 495-499) noted that TAMs express PD-1, and this expression correlates negatively with phagocytic potency against tumor cells. Blockade of PD-1/PD-L1 in vivo increases macrophage phagocytosis, reduces tumor growth and lengthens the survival of mice in cancer models in a macrophage-dependent fashion.

Hsu et al. (Cancer Res 2018;78(22):6349-6353) reported that after exposure to inflammatory cytokines, cancer cells and antigen-presenting cells, such as macrophages and dendritic cells express PD-L1 to inhibit the activity of effector T cells through PD-1 engagement.

As previously mentioned, the main aim of this study was to contribute on the clarification of the prognostic relevance of TAM infiltration in OSCC patients and to establish potential relationships between different components of tumor microenvironment which could be functionally linked and reciprocally regulated to modulate the antitumor immune response and OSCC progression. In this sense, it is undoubtedly the role of tumor PD-L1 expression in immune suppression and tumor progression and in particular the contribution of the CSC niche in OSCC. Nevertheless, we acknowledge the reviewer`s suggestion to further analyze the potential role of PD-L1 expression in macrophages in future studies.

  1. In the method, authors described Disease-specific survival (DSS). However, in table 4 and 5, disease-free survival was showed. They are different. Please clarify

We apologize and thank the reviewer for noticing this inconsistency. Indeed Tables 4 and 5 show DSS not DFS. This has now been corrected and Tables 4 and 5 have been moved to Supplementary Information, according to Reviewer 2’s recommendation.

  1. Except Disease-specific survival and disease-free survival, it is better to show the overall survival

Overall survival has also been analyzed; however, no significant associations with CD68 or CD163 expression were observed. For this reason, overall survival has finally not been included in the text and tables, as it does not provide any relevant information beyond DSS for the survival analysis of this OSCC cohort.

  1. In this manuscript, author showed the mean number of stromal/tumoral CD163+ cells were also higher in tumors with negative expression of NANOG and SOX2. Noteworthy, CD163 + TAM infiltration was inversely and significantly correlated with NANOG expression, in both the stroma (p = 0.001) and tumor nests (p = 0.03). Please described the more detailed rationale to correlated TAM with CSC markers. What is the significance? I think it is more significant if authors can use some in vitro experiments to validate this clinical finding.

We have found an inverse significant relationship between stromal/tumoral CD163+ cells and two CSC markers: NANOG and SOX2. This relationship could suggest a potential role for the CSC niche in immune evasion and OSCC progression, plausibly through paracrine signals to control TAMs infiltration in the tumor microenvironment. In fact, TAMs have been traditionally considered as blood monocytes recruited from the tumor vasculature by tumor-derived signals (Owen JL. Front Physiol 2013;4:159), but more recently it has been shown that most tissue macrophages arise from yolk sac progenitors, with some exceptions such as those from intestines (Hoeffel G, Ginhoux F. Front Immunol 2015;6:486). This fact raises the possibility that TAMs could have a CSC origin in the tumor microenvironment (Osman A, et al. Cancers 2020,12,879).

We herein unveil an unprecedented relationship between TAM infiltration and the oral CSC niche, perhaps reciprocally regulated by paracrine signals. Nevertheless, the precise role of TAMs in regulating CSCs and the molecular mechanisms underlying these interactions in the context of OSCC are not known. In the interplay between CSCs and TAMs a putative role for the signal transducer and activator of transcription 3 (STAT3) pathway has been demonstrated in breast cancer models via a novel paracrine EGFR/Stat3/Sox-2 axis (Yang J, Liao D, Chen C, Liu Y, Chuang TH, Xiang R, Markowitz D, Reisfeld RA, Luo Y. Tumor-associated macrophages regulate murine breast cancer stem cells through a novel paracrine EGFR/Stat3/Sox-2 signaling pathway. Stem Cells 2013;31(2):248-58. doi: 10.1002/stem.1281). In the TME, hypoxia stimulates the expression of interleukin-6 which in turn induces STAT3 in CSCs, and M2 macrophage polarization (Wu A, Wei J, Kong LY, Wang Y, Priebe W, Qiao W, Sawaya R, Heimberger AB. Glioma cancer stem cells induce immunosuppressive macrophages/microglia. Neuro Oncol 2010;12(11):1113-25. doi: 10.1093/neuonc/noq082;  Nusblat LM, Carroll MJ, Roth CM. Crosstalk between M2 macrophages and glioma stem cells. Cell Oncol (Dordr). 2017;40(5):471-482. doi: 10.1007/s13402-017-0337-5). While the ability of CSCs to induce macrophage polarization is well-established, the reciprocal effect of macrophages on the CSC phenotype remains unclear. Nusblat et al. [34] showed that in the presence of conditioned media from M2 macrophages, CSCs increased migration and collagen degradation capacities. In addition, CSCs may exert a strong immune suppressive effect by suppressing the Major Histocompatibility Complex (MHC) by macrophages in the TME, releasing the exosomal miRNAs miR-9 and miR-21, or by secreting VEGF that plays a pivotal role in tumor angiogenesis and suppression of T-cell maturation (Ganguli P, Sarkar RR. Exploring immuno-regulatory mechanisms in the tumor microenvironment: Model and design of protocols for cancer remission. PLoS One. 2018;13(9):e0203030. doi: 10.1371/journal.pone.0203030). Mazzoldi et al. (Mazzoldi EL, Pavan S, Pilotto G, Leone K, Pagotto A, Frezzini S, Nicoletto MO, Amadori A, Pastò A. A juxtacrine/paracrine loop between C-Kit and stem cell factor promotes cancer stem cell survival in epithelial ovarian cancer. Cell Death Dis. 2019;10(6):412. doi: 10.1038/s41419-019-1656-4) found that, the c-Kit ligand Stem Cell Factor (SCF) produced by M2 macrophages, may favor the immune escape of tumor cells in ovarian cancer. Similarly, ovarian CSCs and macrophages reciprocally interact through the WNT/β-catenin signaling thereby contributing to tumorigenesis, invasiveness, and immune suppression (Raghavan S, Mehta P, Xie Y, Lei YL, Mehta G. Ovarian cancer stem cells and macrophages reciprocally interact through the WNT pathway to promote pro-tumoral and malignant phenotypes in 3D engineered microenvironments.J Immunother Cancer. 2019 ;7(1):190. doi: 10.1186/s40425-019-0666-1).

According to these evidences, different cytokines, growth factors and/or mRNAs secreted by the CSCs could mediate a crosstalk between different groups of cells in the tumor microenvironment (TME). According to our findings these regulatory mechanisms that operate in the tumor microenvironment also involve the interplay between the oral CSC niche and TAM infiltration probably contributing to an immune suppressive TME and OSCC progression. This information has now been included in the Discussion.

  1. In table 1, “recurrence” should not be put in table 1. “Recurrence” is time-dependent. It should be presented in disease-free survival using Kaplan-Meier survival curve.

Following the reviewer’s suggestion, recurrence has been eliminated from Table 1.

Reviewer 2 Report

The paper «Macrophages in oral carcinomas: Relationship with cancer stem cell markers and PD-L1 expression» deals with a hot topic, and is of great interest as immune response is important for tumor growth.  The paper is well written, but there are some issues.

In the abstract NANOG is introduced without any explanation. It should be stated that this is a CSC marker.

In Material and methods, patients and tissue specimens the author states that informed consent was obtained form the patients involved in the study. 66% of the patients were dead (line 224) as they were from the period 1996-2007. Was the consent from all the patients alive at study start?
In the same paragraph, the authors give a lot of information about location (very nice that this is stated), how many patients in each T, Stage, differentiation and so on. Does this not belong to results? Line 116-127.

I the result part, 3.2, the authors state that “higher stromal and tumoral CD163+ Infiltrating cells were observed in non-smokers and non-alcoholic drinkers. Similarly, stromal and tumoral CD68+ infiltrating was higher in those patients without tobacco/alcohol habits”. Line 196-198. From table 1 none of these were significant, and the statement is misleading and confusing.  Please rephrase.

Table 4 and 5 do not give any new data, and are not very interesting. Could be omitted, but not mandatory to do so.

Figure 1 and 2: Where is the reference in the text to the figures? Nice pictures – good quality, but I miss in the legends the scoring %. Fig 1A and fig 2A: is this graded as a 50% case?

In the discussion the authors state that there is no association with TAMs and survival, and this just adds up to other reports that the presence of TAM is controversial. This is good that the authors report “negative results”. It is also very good that the authors have clearly separated oral from oropharyngeal tumors. The authors do not have to comment on this!

Author Response

Comments and Suggestions for Authors

The paper «Macrophages in oral carcinomas: Relationship with cancer stem cell markers and PD-L1 expression» deals with a hot topic, and is of great interest as immune response is important for tumor growth.  The paper is well written, but there are some issues.

We thank the reviewer for all the positive comments and for highlighting the interest of our study.

  1. In the abstract NANOG is introduced without any explanation. It should be stated that this is a CSC marker.

This information has been added to the Abstract, as suggested by the reviewer.

  1. In Material and methods, patients and tissue specimens the author states that informed consent was obtained form the patients involved in the study. 66% of the patients were dead (line 224) as they were from the period 1996-2007. Was the consent from all the patients alive at study start?

The consent was obtained at the beginning of the study.

In the same paragraph, the authors give a lot of information about location (very nice that this is stated), how many patients in each T, Stage, differentiation and so on. Does this not belong to results? Line 116-127.

Since these data are all related to the description of the sample, we believe this information has been adequately included in section 2.1 Patients and tissue specimens.

  1. I the result part, 3.2, the authors state that “higher stromal and tumoral CD163+ Infiltrating cells were observed in non-smokers and non-alcoholic drinkers. Similarly, stromal and tumoral CD68+ infiltrating was higher in those patients without tobacco/alcohol habits”. Line 196-198. From table 1 none of these were significant, and the statement is misleading and confusing.  Please rephrase.

We precisely said in the text that higher stromal and tumoral CD163+/ CD68+ infiltrating cells were observed in non-smokers and non-alcoholic drinkers; however, these findings did not reach statistical significance. This means that the mean values for both markers (CD163 and CD68) were higher in non-smokers compared to smokers, and also higher in non-drinkers compared to drinkers.

  1. Table 4 and 5 do not give any new data, and are not very interesting. Could be omitted, but not mandatory to do so.

Following the reviewer’s suggestion, Tables 4 and 5 have been moved to Supplementary Information (now renumbered as Tables S1 and S2 respectively).

  1. Figure 1 and 2: Where is the reference in the text to the figures? Nice pictures – good quality, but I miss in the legends the scoring %. Fig 1ª and fig 2ª: is this graded as a 50% case?

Thanks for this observation. A reference to both figures has now been added to the text. Certainly, Figures 1A and 1B are graded as 50% to each other, as well as Figures 2A and 2B.

  1. In the discussion the authors state that there is no association with TAMs and survival, and this just adds up to other reports that the presence of TAM is controversial. This is good that the authors report “negative results”. It is also very good that the authors have clearly separated oral from oropharyngeal tumors. The authors do not have to comment on this!

Many thanks for your comment. We fully agree.

Round 2

Reviewer 1 Report

Authors have answered my questions. I have no more questions.